# Rethinking Aleatoric and Epistemic Uncertainty

**Freddie Bickford Smith**[1]  **Jannik Kossen**[1]  **Eleanor Trollope**[1]
**Mark van der Wilk**[1]  **Adam Foster**[1]  **Tom Rainforth**[1]

## Abstract

The ideas of aleatoric and epistemic uncertainty are widely used to reason about the probabilistic predictions of machine-learning models. We identify incoherence in existing discussions of these ideas and suggest this stems from the aleatoric-epistemic view being insufficiently expressive to capture all the distinct quantities that researchers are interested in. To address this we present a decision-theoretic perspective that relates rigorous notions of uncertainty, predictive performance and statistical dispersion in data. This serves to support clearer thinking as the field moves forward. Additionally we provide insights into popular information-theoretic quantities, showing they can be poor estimators of what they are often purported to measure, while also explaining how they can still be useful in guiding data acquisition.

## 1 Introduction

When making decisions under uncertainty, it can be useful to reason about where that uncertainty comes from (Osband et al, 2023; Wen et al, 2022). Researchers often aim to do this by referring to aleatoric (literal meaning: "relating to chance") and epistemic ("relating to knowledge") uncertainty, ideas with a long history in the study of probability (Hacking, 1975). Aleatoric uncertainty is typically associated with statistical dispersion in data (sometimes thought of as noise), while epistemic is associated with a model's internal information state (Hüllermeier & Waegeman, 2021).

Concerningly given their scale of use, these ideas are not being discussed coherently in the literature. The line between model-based predictions and data-generating processes is repeatedly blurred (Amini et al, 2020; Ayhan & Berens, 2018; Immer et al, 2021; Kapoor et al, 2022; Smith & Gal, 2018; van Amersfoort et al, 2020). On top of this, tenuous as-

[1]University of Oxford. Correspondence to Freddie Bickford Smith <fbickfordsmith@cs.ox.ac.uk>.

*Proceedings of the 42nd International Conference on Machine Learning*, Vancouver, Canada. PMLR 267, 2025. Copyright 2025 by the author(s).

sumptions are made about how uncertainty will decompose on unseen data (Seeböck et al, 2019; Wang & Aitchison, 2021), and misleading connections are drawn between uncertainty and predictive accuracy (Orlando et al, 2019; Wang et al, 2019). Meanwhile distinct mathematical quantities are used to refer to notionally the same concepts: epistemic uncertainty, for example, has been variously defined using density-based (Mukhoti et al, 2023; Postels et al, 2020), information-based (Gal et al, 2017) and variance-based (Gal, 2016; Kendall & Gal, 2017; McAllister, 2016) quantities.

We suggest this incoherence arises from the aleatoric-epistemic view being too simplistic in the context of machine learning. Researchers are looking for concrete notions of a model's predictive uncertainty and how that uncertainty might or might not change with more data (associated with a decomposition into irreducible and reducible components), but also related notions of predictive performance and data dispersion. The aleatoric-epistemic view cannot satisfy all these needs: many concepts stand to be defined, while the view fundamentally only has capacity for two concepts. Yet the current state of play is to nevertheless appeal to the aleatoric-epistemic view, with different researchers using it in different ways. A result of this conceptual overloading is to conflate quantities that ought to be recognised as distinct. Far from just a matter of semantics, this is having a meaningful effect on the field's progress: methods are being designed and evaluated based on shaky foundations.

To establish a clearer perspective, we draw on powerful yet underappreciated ideas from decision theory (Dawid, 1998; DeGroot, 1962; Neiswanger et al, 2022). Our starting point is a final decision of interest with an associated loss function. Given this, uncertainty in predictive beliefs can be formalised as the subjective expected loss of acting Bayes-optimally under those beliefs; this generalises quantities like variance and Shannon entropy. From there we show how reasoning about new data gives rise to a notion of expected uncertainty reduction, which we can use to identify a decomposition of uncertainty into irreducible and reducible components. Then we clarify the connection between uncertainty, predictive performance and data dispersion, linking to classic decompositions from statistics and information theory. Overall this provides a coherent synthesis of key quantities that researchers are interested in (Figure 1).

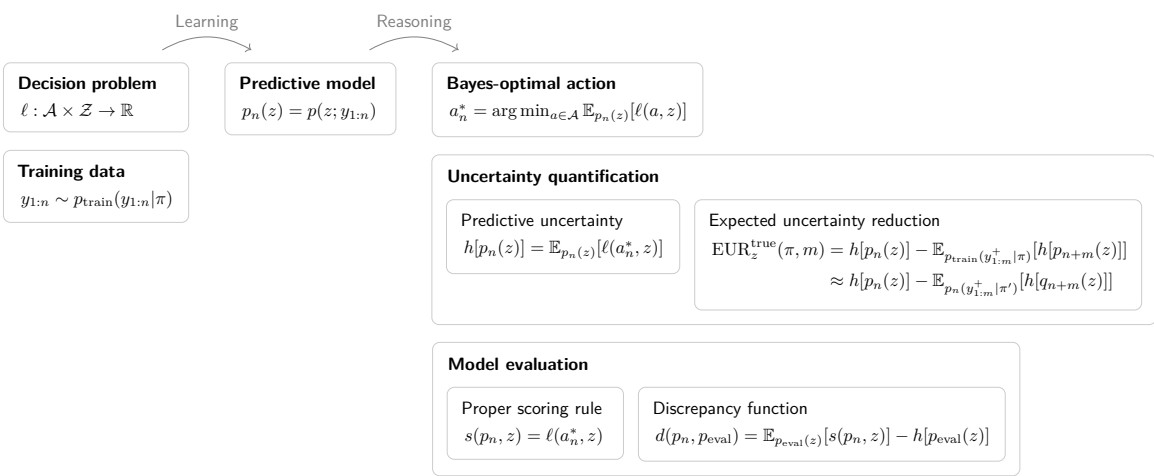

**Figure 1** Our decision-theoretic view coherently relates machine-learning concepts that have been conflated under the aleatoric-epistemic view. We consider taking an action, $a \in \mathcal{A}$, in light of imperfect knowledge of $z \in \mathcal{Z}$, with an action's consequences measured by a loss function, $\ell(a, z)$. Since $z$ is unknown, we use any available training data, $y_{1:n}$, to build a predictive model, $p_n(z)$, with which we can reason over possible values of $z$ and thus choose an action. Additionally we can use the model to quantify uncertainty and its reducibility with respect to new data, and we can evaluate the model using a ground-truth realisation of $z$ or a reference distribution, $p_{\text{eval}}(z)$.

Bridging this generalised perspective back to how aleatoric and epistemic uncertainty have been discussed in past work, we provide new insights on BALD, a popular information-theoretic objective for data acquisition (Gal et al, 2017; Houlsby et al, 2011; Lindley, 1956). In particular we highlight that it should be seen not as a direct measure of long-run reducible predictive uncertainty, as has been suggested in the past, but instead as an estimator that can be highly inaccurate. Reconciling this with BALD's practical utility, we suggest it is often better understood as approximately measuring short-run reductions in parameter uncertainty. It can therefore be useful, albeit still suboptimal in prediction-oriented settings (Bickford Smith et al, 2023; 2024).

Our work thus serves to inform future work in two key ways. On the one hand it sheds light on the contradictions of the aleatoric-epistemic view and presents a coherent alternative perspective that allows clearer thinking about uncertainty in machine learning. On the other hand it provides more direct practical insights. It clarifies that what might have seemed like arbitrary choices for a decision-maker can instead be made by following well-defined logic: given some basic components and principles, it becomes clear how we should measure predictive uncertainty, predictive performance and data dispersion, and how we can identify good future training data. It also highlights approximations that often have to be made in practice, revealing scope for suboptimal performance and therefore informing future methods research.

## 2 Background

The broad motivation of the aleatoric-epistemic view is to distinguish between different sources of uncertainty. If a model's prediction is uncertain, we might want to know

whether that prediction is fundamentally uncertain for the given model class or instead due to a lack of data. This breakdown has clear utility in the context of seeking new data that will reduce predictive uncertainty (Bickford Smith et al, 2023; 2024; MacKay, 1992a;b). But it is also relevant elsewhere: in model selection, for example, we might want to quantify a model's scope for improvement by forecasting how its predictions will change given more data (Barbieri & Berger, 2004; Fong & Holmes, 2020; Geisser & Eddy, 1979; Kadane & Lazar, 2004; Laud & Ibrahim, 1995).

Uncertainty that resolves in light of new data can be thought of as "epistemic" in the sense that data conveys knowledge. Intuitively the corresponding irreducible uncertainty seems to be determined by not only the model class but also, among other things, an "inherent" level of uncertainty associated with the data source at hand, which is often thought of in terms of randomness or chance, hence the word "aleatoric".

While the concepts of aleatoric and epistemic uncertainty had previously been used in machine learning, for example by Lawrence (2012) and Senge et al (2014), their popularity grew following work by Gal (2016), Gal et al (2017) and Kendall & Gal (2017). The most widely used mathematical definitions of these ideas, which we will discuss in Section 4, are the information-theoretic quantities used by Gal et al (2017), building on earlier work on Bayesian experimental design (Lindley, 1956) and Bayesian active learning (Houlsby et al, 2011; MacKay, 1992a;b).

A range of perspectives on aleatoric and epistemic uncertainty in machine learning have been put forward in recent years. These include a discussion of sources of uncertainty in machine learning (Gruber et al, 2023); a case against Shannon entropy as a measure of predictive uncertainty

(Wimmer et al, 2023); proposals for alternative information quantities (Schweighofer et al, 2023a;b; 2025); and various other suggestions for how to define uncertainty, such as in terms of class-wise variance (Sale et al, 2023b; 2024b), credal sets (Hofman et al, 2024a; Sale et al, 2023a), distances between probability distributions (Sale et al, 2024a), frequentist risk (Kotelevskii et al, 2022; 2025; Lahlou et al, 2023) and proper scoring rules (Hofman et al, 2024b). As we will show, our replacement for the aleatoric-epistemic view unifies and explains many of these ideas.

# 3 Key concepts

Our aim in this work is to formalise and link together quantities that have been associated with the ideas of aleatoric and epistemic uncertainty in past work. In particular we look to identify a rigorous notion of predictive uncertainty and the extent to which it reduces as more data is observed, and also measures of predictive performance and statistical dispersion in data. We start by highlighting some foundational concepts that will be used throughout our discussion.

## 3.1 Reasoning should start with the decision of interest

We consider taking an action, $a \in \mathcal{A}$, under imperfect knowledge of a ground-truth variable, $z \in \mathcal{Z}$. Here $z$ could for example be an output relating to a given input (if so, the input is left implicit in our notation) or a parameter in a model, and $a$ could be a direct prediction of $z$, with $\mathcal{A} = \mathcal{Z}$ for point prediction, or $\mathcal{A} = \mathcal{P}(\mathcal{Z})$ for probabilistic prediction. We emphasise our choice to focus on this decision, in deliberate contrast with the more common starting point of learning a model from fixed data. We want a notion of predictive uncertainty that is grounded in actions and their consequences, and we need to reason about different possible datasets to rigorously think about reductions in uncertainty.

## 3.2 Actions induce losses that reflect preferences

We assume we can measure the consequences of taking action $a$ in light of a realisation of $z$ using a loss (or negative utility) function, $\ell : \mathcal{A} \times \mathcal{Z} \to \mathbb{R}$. In principle the specification of $\ell$ follows directly from having preferences that satisfy basic axioms of rationality (von Neumann & Morgenstern, 1947). In practice it can be hard to know what $\ell$ should be; options for dealing with this include using an intrinsic loss or a random loss (Robert, 1996; 2007).

## 3.3 Subjective expected loss enables decision-making

Since $\ell$ is a function of the unknown $z$, it cannot be used directly as an objective for selecting an action, $a$. A principled solution that we focus on here is to form subjective beliefs over possible values of $z$ (conventionally this belief state would be a Bayesian prior or posterior), average over these to form an expected loss, then choose an action

that minimises this subjective expected loss (Ramsey, 1926; Savage, 1951). Alternative decision-making approaches include minimax, which involves minimising the worst-case frequentist risk (von Neumann, 1928; Wald, 1939; 1945).

## 3.4 Machine learning allows data-driven prediction

Minimising subjective expected loss requires beliefs over $z$, and those beliefs can often be informed by some training data, $y_{1:n} \sim p_{\text{train}}(y_{1:n}|\pi)$, where $\pi \in \Pi$ is a policy that controls aspects of data generation. We want notions of uncertainty that reflect how we will actually learn from data, rather than assuming idealised updating that we cannot perform in practice. We therefore define our predictive model, $p_n(z) = p(z; y_{1:n})$, to be the output of a generic machine-learning method applied to the training data (and the input of interest if there is one) for any given $n$, which lets us reason about actual changes in uncertainty as $n$ varies.

Conventional Bayesian inference—taking a generative model over possible data and conditioning on the observed data, giving $p_n(z) = p(z|y_{1:n})$—is one possible updating method. Others include deep learning (LeCun et al, 2015), in-context learning (Brown et al, 2020) and non-Bayesian ensemble methods (Breiman, 2001). In some cases the predictive distribution is defined as $p_n(z) = \mathbb{E}_{p_n(\theta)}[p_n(z|\theta)]$ where $\theta \sim p_n(\theta) = p(\theta; y_{1:n})$ represents a set of stochastic model parameters that we average over at prediction time.

Regardless of whether there are stochastic parameters, the updating scheme could be stochastic. To handle this we take the convention that updating stochasticity is implicitly absorbed into $y_{1:n}$: we can consider our machine-learning method to be a deterministic mapping that takes a random-number seed as an auxiliary input along with data. Thus, while stochastic updating can be source of variability in how uncertainty reduces, this can be dealt with as part of the variability already present in what data we observe.

## 3.5 Bayes optimality is a subjective notion

An action taken by minimising subjective expected loss under $p_n(z)$ is referred to as Bayes optimal (Murphy, 2022); if the action is an estimator of some quantity of interest then it is known as a Bayes estimator. The notion of Bayes optimality assumes our beliefs, $p_n(z)$, represent our best knowledge of $z$, and $\ell$ reflects our preferences. It says nothing at all about how well our beliefs match reality, or about the actions we would take if we had different beliefs. Bayes-optimal actions can therefore be suboptimal as judged using realisations of $z$ from somewhere other than $p_n(z)$, such as a system serving as a source of ground truth.

## 3.6 Predictions often do not match data generation

The correspondence between our predictions, $p_n(z)$, and the data-generating process, $p_{\text{train}}(y_i|\pi(y_{<i}), y_{<i})$, can be

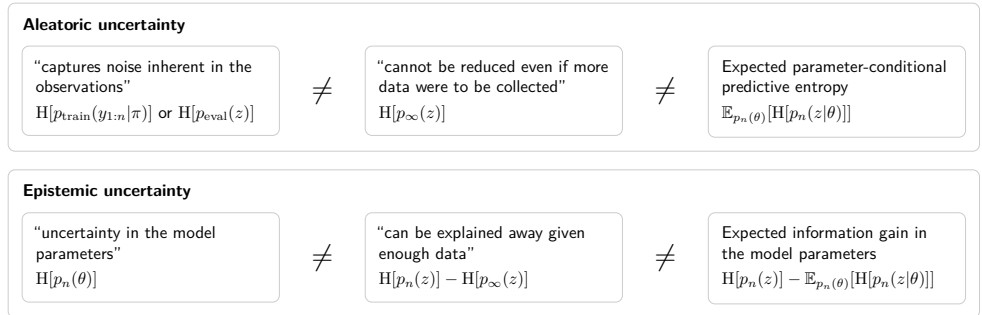

**Figure 2** A popular interpretation of aleatoric and epistemic uncertainty in machine learning attaches multiple mathematical quantities to each of the two concepts, rendering it incoherent and thus a likely source of conflations in the literature. Here we show quotations from Kendall & Gal (2017) expressed mathematically as information-theoretic quantities, along with an interpretation of Equation 1 due to Gal (2016) and Gal et al (2017). Some of the quantities shown can coincide in particular cases but in the general case they are distinct.

weak. One basic reason for this is that they might be defined over different event spaces. We could for example have $y_i \notin \mathcal{Z}$: perhaps we want to predict a coin's bias based on outcomes of coin tosses, or we want to predict a variable in one domain (eg, vision) based on data from another domain (eg, text). Even if that is not the case, $p_n(z)$ is a reflection of assumptions and design decisions based on incomplete knowledge (Box, 1976; Kleijn & van der Vaart, 2006), and there is no general guarantee that it will match reality.

### 3.7 Model evaluation relies on external grounding

Because we expect $p_n(z)$ to be imperfect, we often want to assess it against a ground-truth realisation of $z$ or a reference distribution, $p_{\text{eval}}(z)$. Commonly this distribution—which could for example represent a computer program, a human expert or a physical sensor—is used as source of evaluation data for computing an estimator of frequentist risk (Berger, 1985), such as the mean squared error. Notably $p_{\text{eval}}(z)$ could itself be imperfect (Fluri et al, 2023), so designing and interpreting evaluations of this form requires care.

## 4 Assessing a popular view

Having outlined intuitive descriptions of aleatoric and epistemic uncertainty and their motivation in Section 2, we turn to how they have been formalised in machine learning. Aleatoric and epistemic uncertainty are often thought of as additive components of predictive uncertainty. A popular way to formalise this for models with stochastic parameters, $\theta$, is to relate three information-theoretic quantities:

$$\underbrace{\text{EIG}_\theta}_{\text{"epistemic"}} = \underbrace{\text{H}[p_n(z)]}_{\text{"total"}} - \underbrace{\mathbb{E}_{p_n(\theta)}[\text{H}[p_n(z|\theta)]]}_{\text{"aleatoric"}} \quad (1)$$

where H denotes Shannon entropy and $\text{EIG}_\theta$, also known as the BALD score (Houlsby et al, 2011), is the expected information gain in $\theta$ from observing $z$. Gal (2016) stated the "total = aleatoric + epistemic" relationship and the correspondence between $p_n(z)$ and total uncertainty, while Gal

et al (2017) made the explicit link to Equation 1, informed by Houlsby et al (2011). Kendall & Gal (2017) discussed the aleatoric-epistemic view in the context of computer vision.

While that work successfully captured some of the intuitions from Section 2, we highlight that it also overloaded the ideas of aleatoric and epistemic uncertainty with multiple meanings, introducing a number of spurious associations (Figure 2). The competing definitions of aleatoric uncertainty conflate $\text{H}[p_\infty(z)]$, the entropy of $p_n(z)$ as $n \to \infty$ (this depends on the data-generating process, as we will discuss in Section 5.2), with three separate quantities:

(a) $\text{H}[p_{\text{train}}(y_{1:n}|\pi)]$, the entropy in training-data generation. Issue: $p_\infty(z)$ represents subjective predictive beliefs that need not match $p_{\text{train}}(y_{1:n}|\pi)$ (Section 3.6).

(b) $\text{H}[p_{\text{eval}}(z)]$, the entropy in evaluation-data generation. Issue: $p_\infty(z)$ represents subjective predictive beliefs that need not match $p_{\text{eval}}(z)$ (Sections 3.6 and 3.7).

(c) $\mathbb{E}_{p_n(\theta)}[\text{H}[p_n(z|\theta)]]$, the expected conditional predictive entropy. Issue: for finite $n$ the expected conditional predictive entropy is only an estimator of $\text{H}[p_\infty(z)]$, and it can be highly inaccurate (Section 5.5).

Meanwhile the multiple definitions of epistemic uncertainty mix up $\text{H}[p_n(z)] - \text{H}[p_\infty(z)]$, the predictive-entropy reduction from updating on infinite new data, with two quantities:

(a) $\text{H}[p_n(\theta)]$, the entropy of the model's stochastic parameters. Issue: the mapping from parameters to predictions is typically not invertible, so $\text{H}[p_n(\theta)]$ will not necessarily relate to the reduction in predictive entropy.

(b) $\text{H}[p_n(z)] - \mathbb{E}_{p_n(\theta)}[\text{H}[p_n(z|\theta)]]$, the expected information gain in the model parameters. Issue: for finite $n$ this EIG is only an estimator of $\text{H}[p_n(z)] - \text{H}[p_\infty(z)]$, and the estimation error can be large (Section 5.5).

Other sources of confusion in this aleatoric-epistemic view include an incorrect association between a model's subjective uncertainty and frequentist measures of performance,

such as classification accuracy (Figure 2 in Kendall & Gal (2017)), along with misleading implications about how a model's uncertainty will behave with varying $n$ (Figure 6.11-6.12 in Gal (2016) and Table 3 in Kendall & Gal (2017)) and varying distance from the training data ("Aleatoric uncertainty does not increase for out-of-data examples... whereas epistemic uncertainty does" in Kendall & Gal (2017)).

## 5   An alternative perspective

We now present a coherent, general synthesis of key ideas used in existing discussions of aleatoric and epistemic uncertainty (Figure 1). We begin by reasoning about the subjective expected loss of acting Bayes-optimally under a given belief state, which leads to a decision-grounded measure of predictive uncertainty. By thinking about how that uncertainty will change in light of new data, we then identify a notion of expected uncertainty reduction, which we use to define a decomposition of uncertainty into irreducible and reducible components, and which also has direct practical relevance. Then, shifting our focus to externally grounded evaluation, we highlight the distinction between uncertainty, predictive performance and data dispersion. Finally we return to the BALD score discussed in Section 4, providing insights on its utility as a data-acquisition objective.

### 5.1   Predictive uncertainty can be derived from the final decision of interest and the associated loss

We first deal with the question of how to measure predictive uncertainty, to which many different answers have been put forward (Sections 1 and 2). Revisiting past work, we show that minimising subjective expected loss (Ramsey, 1926; Savage, 1951) directly leads to a loss-grounded measure of uncertainty that reflects our preferences about model behaviour in the final decision of interest. We thus clarify that a decision-maker does not face an arbitrary choice over uncertainty measures: if they specify a loss function based on their preferences, a rigorous uncertainty measure follows.

If $p_n(z)$ represents our beliefs over $z$ then we can identify the Bayes-optimal action, $a_n^*$, in our final decision of interest by minimising the expected loss under those beliefs:

$$a_n^* = \arg\min_{a \in \mathcal{A}} \mathbb{E}_{p_n(z)}[\ell(a, z)]. \tag{2}$$

Now we can reason about the loss we expect (under our belief state) to incur by taking this Bayes-optimal action. An important, underappreciated result is that this minimal expected loss provides a way to measure uncertainty in $p_n(z)$ (Dawid, 1998; DeGroot, 1962; Neiswanger et al, 2022):

$$h[p_n(z)] = \mathbb{E}_{p_n(z)}[\ell(a_n^*, z)].$$

A crucial implication of this is that any two decision-makers should not necessarily use the same uncertainty measure, depending on their decisions of interest and loss functions. One might use variance (Hastie et al, 2009) while the other uses entropy (Shannon, 1948), as Examples 1 and 2 show.

**Example 1** (Dawid, 1998) *Point prediction with $\mathcal{A} = \mathcal{Z}$ and $\ell(a, z) = (a - z)^2$ corresponds to measuring uncertainty in our beliefs, $p_n(z)$, using variance.*

*Proof*   The optimal action is the mean of $p_n(z)$:

$$a_n^* = \arg\min_{a \in \mathcal{A}} \mathbb{E}_{p_n(z)}\big[(a - z)^2\big] = \mathbb{E}_{p_n(z)}[z].$$

The subjective expected loss of taking this action is the variance of $p_n(z)$:

$$h[p_n(z)] = \mathbb{E}_{p_n(z)}\big[(\mathbb{E}_{p_n(z)}[z] - z)^2\big] = \mathbb{V}_{p_n(z)}[z]. \quad \square$$

**Example 2** (Dawid, 1998) *Probabilistic prediction with $\mathcal{A} = \mathcal{P}(\mathcal{Z})$ and $\ell(a, z) = -\log a(z)$ corresponds to measuring uncertainty in our beliefs, $p_n(z)$, using entropy.*

*Proof*   The optimal action is $p_n(z)$:

$$a_n^* = \arg\min_{a \in \mathcal{A}} -\mathbb{E}_{p_n(z)}[\log a(z)] = p_n(z).$$

The subjective expected loss of taking this action is the Shannon entropy of $p_n(z)$:

$$h[p_n(z)] = -\mathbb{E}_{p_n(z)}[\log p_n(z)] = \mathrm{H}[p_n(z)]. \quad \square$$

### 5.2   Decomposing predictive uncertainty requires accounting for the data-generating process

Next, to formalise the popular idea of decomposing predictive uncertainty into irreducible and reducible components (Sections 2 and 4), we reason about how predictive uncertainty changes in light of new data. We show that, as long as we explicitly account for the data-gathering process, we can write down a notion of expected uncertainty reduction that is well defined for any method that maps from data to a predictive distribution (Section 3.4). From this we identify a rigorous irreducible-reducible decomposition.

Characterising the reducibility of uncertainty initially seems as simple as considering new data, $y_{1:m}^+ = y_{(n+1):(n+m)}$, and measuring the corresponding uncertainty reduction,

$$\mathrm{UR}_z(y_{1:m}^+) = h[p_n(z)] - h[p_{n+m}(z)],$$

which we note could be negative. But this uncertainty reduction depends on exactly what the new data is, and that in turn depends on the process by which the data is generated. We therefore need to explicitly account for the data-generating process to produce a well-defined notion of reducibility. Any stochasticity in model updating also has to be taken into account, but this can be achieved by absorbing the updating stochasticity into the definition of $y_{1:m}^+$ (Section 3.4).

Revisiting the data-generating process from Section 3.4, we define the distribution over $y_i^+$ to depend on decisions made by the data-acquisition policy, $\pi$, and on the previous data:

$$p_{\text{train}}(y_{1:m}^+|\pi) = \prod_{i=1}^{m} p_{\text{train}}(y_i^+|\pi(y_{<i}), y_{<i}).$$

With this we can define the true expected uncertainty reduction (EUR) in $z$ under a given policy, $\pi$, as

$$\text{EUR}_z^{\text{true}}(\pi, m) = \mathbb{E}_{p_{\text{train}}(y_{1:m}^+|\pi)}[\text{UR}_z(y_{1:m}^+)]. \quad (3)$$

This allows us to shift from thinking about a specific realisation of data to the range of possible data that might be generated. Working from this EUR to an uncertainty decomposition, we consider the limit of $m \to \infty$:

$$\underbrace{\text{EUR}_z^{\text{true}}(\pi, \infty)}_{\text{reducible}} = \underbrace{h[p_n(z)]}_{\text{total}} - \underbrace{\mathbb{E}_{p_{\text{train}}(y_{1:\infty}^+|\pi)}[h[p_\infty(z)]]}_{\text{irreducible}}.$$

Thus we see that three components—a loss function, a machine-learning method mapping from data to a predictive distribution, and a data-acquisition policy—fully specify a rigorous measure of expected uncertainty reduction and an associated irreducible-reducible decomposition. This contrasts with the decomposition in Equation 1, which requires stochastic model parameters and exact Bayesian updating.

It is worth noting that in some restricted cases the dependency on the data-acquisition policy, $\pi$, of the infinite-data terms in this uncertainty decomposition can disappear. For example, if we are in a supervised-learning setting where the policy's decisions concern which inputs to acquire labels for, and if we are using a well-specified Bayesian model and exact Bayesian updating, $p_\infty(z)$ should be independent of $\pi$ as long as $\pi$ produces dense samples across the input space (Kleijn & van der Vaart, 2012). However, the requirements for this are very strict, with any model misspecification or error in belief updating reintroducing the dependency.

## 5.3 Practical estimation of expected uncertainty reduction relies on approximations

Now we turn to estimating expected uncertainty reduction (EUR) in practice. The decomposition in Section 5.2 is well defined but the infinite-data quantities within it are typically not practically obtainable. While this might seem problematic, we suggest the takeaway should in fact be to deemphasise the decomposition in the context of real-world machine learning, where we do not have infinite data. There is more concrete value (eg, for data acquisition or model selection) in estimating the EUR in Equation 3 for finite $m$, which relies on some important practical approximations.

Since we typically do not know the true data-generating process, $p_{\text{train}}(y_{1:m}^+|\pi)$, a core approximation is to use a model over new data, $p_n(y_{1:m}^+|\pi')$, as a proxy. On top of

this we might also need to approximate how our beliefs over $z$ update when we obtain new data. In principle the EUR is defined with respect to whichever updating scheme we are using, but in practice the true update can be too expensive to perform within an expectation over new data. This can be addressed by using some $q_{n+m}(z)$ in place of $p_{n+m}(z)$. A common approach is to assume a Bayesian belief update given $y_{1:m}^+$, even if the true update is not Bayesian (Bickford Smith et al, 2023; 2024; Gal et al, 2017; Kirsch et al, 2019; 2023). It might be, for example, that we want to use a Bayesian model but cannot perform exact inference. If the true update is Bayesian and can be performed exactly then this assumption is of course not an approximation.

Combining model-based data simulation with an approximate updating scheme, we can estimate the true EUR using

$$\text{EUR}_z^{\text{est}}(\pi', m) = h[p_n(z)] - \mathbb{E}_{p_n(y_{1:m}^+|\pi')}[h[q_{n+m}(z)]].$$

The accuracy of this estimator depends on the mismatch between $p_n(y_{1:m}^+|\pi')$ and $p_{\text{train}}(y_{1:m}^+|\pi)$ as well as the mismatch between $q_{n+m}(z)$ and $p_{n+m}(z)$, both of which are likely to be greater for larger $m$. Estimation therefore requires careful tradeoffs to mitigate these mismatches.

The practical relevance of this becomes clearer upon appreciating that the EUR estimator generalises a number of existing data-acquisition objectives. Under the exact-Bayesian-updating assumption it is equivalent to what has variously been called the "expected value of [additional/sample] information" (Bernardo & Smith, 1994; Raiffa & Schlaifer, 1961), the "expected $H_{\ell,\mathcal{A}}$-information gain" (Neiswanger et al, 2022) and the "expected decision utility gain" (Huang et al, 2024). From that objective we can then recover the expected information gain in $z$, which corresponds to the BALD score (Gal et al, 2017; Houlsby et al, 2011) if $z = \theta$ represents a set of stochastic model parameters, or the expected predictive information gain (Bickford Smith et al, 2023) if $z = (x_*, y_*)$ represents a target input and its output. Notably $\text{EUR}_z^{\text{est}}(\pi', m)$ is also closely related to the idea of the martingale posterior (Fong et al, 2023) as $p_n(y_{1:m}^+|\pi')$ and our updating scheme together imply a joint distribution from which a martingale posterior can be derived.

## 5.4 Model-based uncertainty should be used with care, and externally grounded evaluation is crucial

Next we clarify the relationship between the predictive uncertainty we have discussed so far and quantities commonly associated with it (Sections 2 and 4): measures of predictive performance and data dispersion. We identify how a model could be used to estimate those quantities but emphasise the limitations of that approach, underlining the key role to be played by externally grounded evaluation (Section 3.7).

We consider assessing a predictive distribution, $p_n(z)$, either using a single ground-truth value of $z$ or using a reference

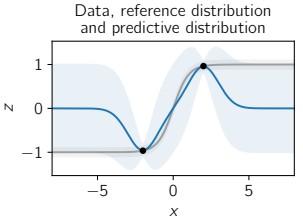 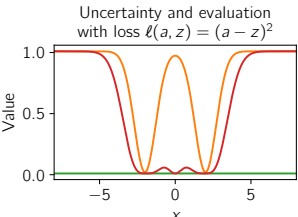 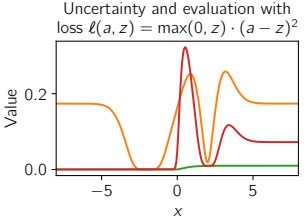 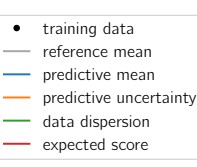

**Figure 3** Model-based uncertainty, predictive performance and data dispersion are related but distinct quantities. Here we show a model's predictive distribution over an output, $z \in \mathbb{R}$, corresponding to an input, $x \in \mathbb{R}$; we also show a reference distribution serving as a source of evaluation data. Using the model alone, we compute a measure of predictive uncertainty using a loss function, $\ell(a, z)$, where $a$ is an action. This differs from the data dispersion, which describes the reference distribution. Connecting the two distributions, the expected score (lower is better) measures the predictive performance of the model as judged using data from the reference distribution.

distribution over $z$. If we have a single $z$, we can evaluate $p_n(z)$ using a proper scoring rule (Savage, 1971) of the form

$$s(p_n, z) = \ell(a_n^*, z)$$

where $a_n^*$ is defined as in Equation 2 (Dawid, 1998). This scoring rule measures the loss incurred by the Bayes-optimal action under $p_n(z)$ when the ground truth is $z$.

If we instead have a reference distribution, $p_{\text{eval}}(z)$, we can evaluate $p_n(z)$ using a discrepancy function (Dawid, 1998):

$$d(p_n, p_{\text{eval}}) = \mathbb{E}_{p_{\text{eval}}(z)}[\ell(a_n^*, z) - \ell(a_{\text{eval}}^*, z)] \quad (4)$$

$$= \underbrace{\mathbb{E}_{p_{\text{eval}}(z)}[s(p_n, z)]}_{\text{expected score}} - \underbrace{h[p_{\text{eval}}(z)]}_{\text{data dispersion}} \quad (5)$$

where $a_{\text{eval}}^*$ is defined analogously to $a_n^*$ but with $p_{\text{eval}}(z)$ as the predictive distribution. Equation 4 highlights that the discrepancy is the expected excess loss from acting based on $p_n(z)$ rather than $p_{\text{eval}}(z)$ when $p_{\text{eval}}(z)$ represents the ground truth, which is linked to the idea of regret (Szepesvári, 2010). Equation 5 shows that the discrepancy also corresponds to the expected score of $p_n(z)$, which measures predictive performance (lower is better), minus the uncertainty measure from Section 5.1 applied to $p_{\text{eval}}(z)$, which measures the statistical dispersion in evaluation data drawn from the reference distribution. This relationship generalises classic decompositions from statistics (Rice, 2007) and information theory (Cover & Thomas, 2005), as shown in Examples 3 and 4. Notably the expected score can be seen as a form of frequentist risk for an "outer" decision problem; another form also averages over the training data.

Because past work has sometimes drawn connections between model-based uncertainty, predictive performance and data dispersion, we now explain how such connections can come about and why in the general case they should not be taken to hold. In particular, if we assume a specific model setup and estimation loss then we can derive Bayes estimators that generalise the marginal predictive entropy and the expected conditional predictive entropy in Equation 1, but these assumptions will typically not apply and, even if they do apply, the estimators can be highly inaccurate.

**Proposition 1** (Berger, 1985) *Let $F$ be a quantity of interest, and let $f(\theta)$ represent subjective beliefs over $F$, derived from a pushforward of a distribution on model parameters $\theta \sim p_n(\theta)$. Under a quadratic estimation loss, $\ell_\eta(\eta, \theta) = (\eta - f(\theta))^2$, the Bayes estimator of $F$ is $\eta^* = \mathbb{E}_{p_n(\theta)}[f(\theta)]$.*

**Proposition 2** *Assume $p_n(z) = \mathbb{E}_{p_n(\theta)}[p_n(z|\theta)]$ is a model intended to directly approximate $p_{\text{eval}}(z)$. Then the model's predictive uncertainty, $h[p_n(z)]$, is a Bayes estimator of $\mathbb{E}_{p_{\text{eval}}(z)}[s(p_n, z)]$, the expected loss from acting Bayes-optimally under $p_n(z)$ when $z$ is in fact drawn from $p_{\text{eval}}(z)$.*

*Proof* Applying Proposition 1 with $F = \mathbb{E}_{p_{\text{eval}}(z)}[s(p_n, z)]$ and $f(\theta) = \mathbb{E}_{p_n(z|\theta)}[s(p_n, z)]$ gives $\eta^* = h[p_n(z)]$. $\square$

**Proposition 3** *Assume the same model as in Proposition 2. Then the expected conditional predictive uncertainty, $\mathbb{E}_{p_n(\theta)}[h[p_n(z|\theta)]]$, is a Bayes estimator of $h[p_{\text{eval}}(z)]$, the dispersion in data drawn from a reference distribution.*

*Proof* Applying Proposition 1 with $F = h[p_{\text{eval}}(z)]$ and $f(\theta) = h[p_n(z|\theta)]$ gives $\eta^* = \mathbb{E}_{p_n(\theta)}[h[p_n(z|\theta)]]$. $\square$

Three things are important to note in regard to Propositions 2 and 3. First, while they both suppose that $p_n(z)$ is designed to directly approximate $p_{\text{eval}}(z)$, this need not always be the case. Because $p_{\text{eval}}(z)$ might itself be imperfect, our evaluation might just be serving to provide a rough signal of the model's predictive performance (Section 3.7). Second, they both assume a quadratic estimation loss, which might not reflect our preferences. An alternative loss would result in different Bayes estimators. For example, an absolute loss would lead us to use medians rather than expectations (Berger, 1985). Third, they present estimators that are derived from subjective models and that, as a result, have no accuracy guarantee in the general case (Section 3.5).

We therefore stress that model-based uncertainty should be considered separate from predictive performance and data dispersion, as shown in Figure 3 (see Appendix C for details). Uncertainty alone is not a reliable indicator of whether we can trust a model. Some kind of external grounding is crucial for well-informed practical deployment.

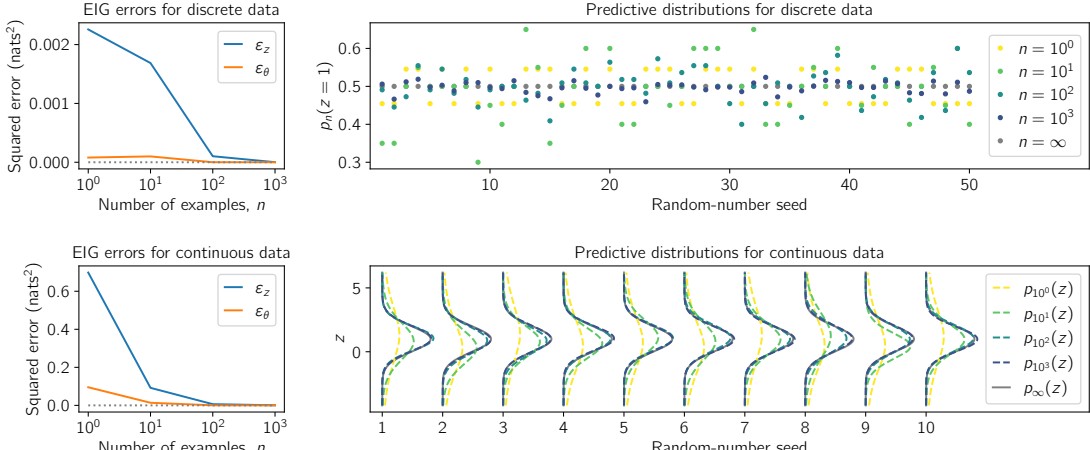

**Figure 4** BALD's correspondence with infinite-step predictive information gain (a model's long-run reduction in predictive entropy) can be weak, such that it is often better thought of as an estimator of a "true" one-step expected information gain in the model parameters. Here we show the behaviour of conjugate models trained on discrete (top) and continuous data (bottom). For $n \in (1, 10, 100, 1000)$ we computed two estimation errors: $\varepsilon_z = (\mathrm{EIG}_\theta - \mathrm{IG}_z(y^+_{1:\infty}))^2$ and $\varepsilon_\theta = (\mathrm{EIG}_\theta - \mathrm{EIG}_\theta^{\mathrm{true}})^2$. We show these errors (left; mean over 50 random-number seeds) along with the evolution of the predictive distribution (right). Both estimation errors are due to inaccurately simulating future data, an issue that resolves as $n$ increases and the predictive distribution converges to the data-generating process.

## 5.5 Popular information-theoretic quantities are best understood as imperfect estimators

Finally we return to the information-theoretic quantities in Equation 1, which have been central to many existing discussions of aleatoric and epistemic uncertainty. Principally we highlight that BALD (that is, $\mathrm{EIG}_\theta$, the expected information gain in a set of stochastic model parameters, $\theta$) can be understood as an estimator of two separate unknown quantities: the infinite-step information gain in the model predictions and a "true" one-step expected information gain in the model parameters. We also suggest the relative magnitudes of the two corresponding estimation errors might help explain BALD's utility as a data-acquisition objective.

First we show that, under assumptions on the data and model that result in convergence to a single setting of $\theta$ (Doob, 1949; Freedman, 1963; 1965), BALD can be understood as an estimator of the infinite-step predictive information gain,

$$\mathrm{IG}_z(y^+_{1:\infty}) = \mathrm{H}[p_n(z)] - \mathrm{H}[p_n(z|y^+_{1:\infty})],$$

measuring the reduction in the model's predictive entropy from a Bayesian update on infinite new data, $y^+_{1:\infty}$.

**Proposition 4** *Let $y^+_{1:m}$ and $p_n(y|\theta)p_n(z|\theta)p_n(\theta)$ be a combination of data sequence and generative model that yield $p_n(\theta|y^+_{1:m}) \to \delta_{\theta_\infty}(\theta)$ as $m \to \infty$. Then the expected conditional predictive entropy, $\mathbb{E}_{p_n(\theta)}[\mathrm{H}[p_n(z|\theta)]]$, is a Bayes estimator of $\mathrm{H}[p_n(z|y^+_{1:\infty})]$, the marginal predictive entropy after a Bayesian update on infinite new data, $y^+_{1:\infty}$.*

**Proposition 5** *Assume the data and model from Proposition 4. Then the expected information gain in the model*

*parameters, $\mathrm{EIG}_\theta$, from observing $z$ is a Bayes estimator of the infinite-step predictive information gain, $\mathrm{IG}_z(y^+_{1:\infty})$.*

In Figure 4 we demonstrate that the approximation $\mathrm{EIG}_\theta \approx \mathrm{IG}_z(y^+_{1:\infty})$ from Proposition 5 can be coarse. These results were produced with extremely simple setups within which we can perform exact inference and we are sure to recover the true data-generating process in the limit of infinite data (see Appendix C for details). We therefore know that the estimation error is due to a failure of the model to accurately simulate future data, which in turn is due to $n$ being finite.

This behaviour appears to align with existing results (with the caveat that past studies did not match the assumptions of Propositions 4 and 5). Figure 2 in Bickford Smith et al (2024) and Figure 5 in Wimmer et al (2023) show small-$n$ estimates of $\mathrm{EIG}_\theta$ that differ substantially from the changes in predictive entropy that actually occurred in practice. Those results even suggest it would have been more accurate to assume $\mathrm{IG}_z(y^+_{1:\infty}) = \mathrm{H}[p_n(z)]$, with $\mathrm{H}[p_n(z|y^+_{1:\infty})] = 0$, than to estimate it using $\mathrm{EIG}_\theta$. Meanwhile Mucsányi et al (2024) and Valdenegro-Toro & Saromo-Mori (2022) emphasised the "entanglement" of aleatoric- and epistemic-uncertainty estimators, which can be understood as the estimators themselves having an "epistemic" component—or, in our terminology, being inaccurate finite-data estimators.

This raises the question of why BALD has proven practically useful as a data-acquisition objective in active learning (Gal et al, 2017; Houlsby et al, 2011; Osband et al, 2023). If our intuition is that BALD's utility stems from its correspondence with infinite-step predictive information gain (that is, a long-run reduction in a model's predictive entropy) and

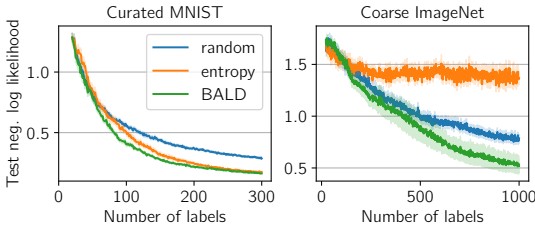

**Figure 5** BALD outperforms predictive entropy as a data-acquisition objective in active learning, even though BALD tends to be a worse estimator of long-run predictive information gain in the setups studied. These results were produced using experimental setups described in Bickford Smith et al (2023; 2024).

we consider setups in which the current predictive entropy is a better estimator than BALD, then we would expect that using the predictive entropy as a data-acquisition objective would lead to better predictive performance in active learning. Yet this is not what we see in practice (Figure 5).

We suggest another perspective on BALD might address this question. In particular it can be understood (given assumptions on the data and model) as an estimator of the true one-step expected information gain in the model parameters,

$$\text{EIG}_\theta^{\text{true}} = \text{H}[p_n(\theta)] - \mathbb{E}_{p_{\text{train}}(z)}[\text{H}[p_n(\theta|z)]],$$

where the expectation is over observations from the true data-generating process, not model-simulated observations.

**Proposition 6** *Let $z = y_{n+1}$. Assume $y_{1:n}$ are independent and identically distributed, with $y_i \sim p_{\text{train}}(y)$, and assume $p_n(z) = \mathbb{E}_{p_n(\theta)}[p_n(z|\theta)]$ is a model intended to directly approximate $p_{\text{train}}(z)$. Then the expected information gain in the model parameters, $\text{EIG}_\theta$, from observing $z$ is a Bayes estimator of the true one-step expected information gain, $\text{EIG}_\theta^{\text{true}}$, where the expectation is with respect to $p_{\text{train}}(z)$.*

Returning to the same experimental setup as before, we find (Figure 4) that the approximation $\text{EIG}_\theta \approx \text{EIG}_\theta^{\text{true}}$ is more accurate than $\text{EIG}_\theta \approx \text{IG}_z(y_{1:\infty}^+)$. In other words, BALD more closely tracks short-run changes in parameter uncertainty than it does long-run changes in predictive uncertainty. We do not claim this is a general result that will hold in all settings, but it is consistent with BALD being useful as a data-acquisition objective. The data-acquisition horizons in active learning are typically very short, so it is the short-run notion of information gain that matters, not the asymptotic notion. And while targeting predictions rather than parameters can be even more effective (Bickford Smith et al, 2023; 2024), maximising short-run parameter information gain is still often preferable over random acquisition.

One takeaway here is that information-theoretic quantities should not be confused with the quantities they estimate. Another is that expected information gain in a variable of interest is a well-motivated objective for data acquisition, assuming the action of interest is a probabilistic prediction of $z$ and we use a negative-log-likelihood loss function (Example 2), but it also crucially depends on the model's ability to simulate future data, and it assumes a Bayesian update that might not match the true update (Section 5.3).

## 6 Conclusion

We have argued that the aleatoric-epistemic view on uncertainty does not serve machine-learning researchers' needs: its lack of expressive capacity has led to conceptual overloading and confusion. To address this we have presented a decision-theoretic view that unifies many concepts of interest to researchers. This provides clarity on five key points:

(a) Measures of predictive uncertainty need not be an arbitrary choice but can instead be derived from a decision of interest with an associated loss function.

(b) If we explicitly account for how training data is generated, we can identify a decomposition of uncertainty into reducible and irreducible components for any method that maps from data to a predictive distribution.

(c) In practice we can typically only produce an approximate notion of expected uncertainty reduction that relies on a proxy for the true data-generating process and possibly also an approximation of model updating.

(d) Predictive uncertainty should be assumed to be separate from measures of predictive performance and data dispersion, and externally grounded evaluation is therefore key for building trust in a model's predictions.

(e) BALD does not directly measure long-run reducible predictive uncertainty but rather estimates it, and the associated estimation error can be large.

We believe our decision-theoretic view should be used in place of the predominant aleatoric-epistemic view as the field moves forward. Our hope is that it will support more productive discourse and methodological development.

We also urge care in using uncertainty and related quantities in practice. A crucial recurring point in this work is that true quantities of interest almost always have to be approximated in the real world. Ignoring approximation errors can lead to dangerous assumptions and ineffective methods.

Finally we suggest that future methods research should be guided by practical utility in concrete decision problems, such as data acquisition and model selection, rather than more abstract notions of approximation accuracy. Combined with stronger conceptual foundations, a renewed emphasis on real-world performance could accelerate progress in the use of probabilistic reasoning in machine learning.

## Impact statement

This paper presents work whose goal is to advance the field of machine learning. There are many potential societal consequences of our work, none which we feel must be specifically highlighted here.

## Acknowledgements

We thank Philip Dawid for sharing a technical report with us. We are also grateful to Andreas Kirsch, Mike Osborne and the anonymous reviewers of this paper for useful discussions and feedback. Freddie Bickford Smith is supported by the EPSRC Centre for Doctoral Training in Autonomous Intelligent Machines and Systems (EP/L015897/1). Tom Rainforth is supported by EPSRC grant EP/Y037200/1.

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

# A  Examples

**Example 3** *Evaluating $p_n(z)$ against a reference distribution, $p_{\text{eval}}(z)$, with $\mathcal{A} = \mathcal{Z}$ and $\ell(a, z) = (a - z)^2$ corresponds to measuring the predictive performance of $p_n(z)$ using mean squared error, measuring the discrepancy between $p_n(z)$ and $p_{\text{eval}}(z)$ using squared bias and measuring dispersion in $p_{\text{eval}}(z)$ using variance.*

*Proof* Starting from Equation 4 with optimal actions $a_n^* = \mathbb{E}_{p_n(z)}[z] = \mu_n$ and $a_{\text{eval}}^* = \mathbb{E}_{p_{\text{eval}}(z)}[z] = \mu_{\text{eval}}$ from Example 1, the discrepancy between $p_n(z)$ and $p_{\text{eval}}(z)$ is

$$d(p_n, p_{\text{eval}}) = \mathbb{E}_{p_{\text{eval}}(z)}\big[(\mu_n - z)^2 - (\mu_{\text{eval}} - z)^2\big] = (\mu_n - \mu_{\text{eval}})^2.$$

Now starting from Equation 5, it can also be written as

$$d(p_n, p_{\text{eval}}) = \mathbb{E}_{p_{\text{eval}}(z)}\big[(\mu_n - z)^2\big] - \mathbb{V}_{p_{\text{eval}}(z)}[z].$$

Equating these two expressions for the discrepancy leads to a standard bias-variance decomposition:

$$\underbrace{\mathbb{E}_{p_{\text{eval}}(z)}\big[(\mu_n - z)^2\big]}_{\text{mean squared error}} = \underbrace{(\mu_n - \mu_{\text{eval}})^2}_{\text{squared bias}} + \underbrace{\mathbb{V}_{p_{\text{eval}}(z)}[z]}_{\text{variance}}$$

where the mean squared error measures predictive performance and the variance measures data dispersion. $\square$

**Example 4** *Evaluating $p_n(z)$ against a reference distribution, $p_{\text{eval}}(z)$, with $\mathcal{A} = \mathcal{P}(\mathcal{Z})$ and $\ell(a, z) = -\log a(z)$ corresponds to measuring the predictive performance of $p_n(z)$ using cross entropy, measuring discrepancy between $p_n(z)$ and $p_{\text{eval}}(z)$ using Kullback-Leibler divergence and measuring dispersion in $p_{\text{eval}}(z)$ using Shannon entropy.*

*Proof* Starting from Equation 4 with optimal actions $a_n^* = p_n(z)$ and $a_{\text{eval}}^* = p_{\text{eval}}(z)$ (Example 2), the discrepancy between $p_n(z)$ and $p_{\text{eval}}(z)$ is

$$d(p_n, p_{\text{eval}}) = \mathbb{E}_{p_{\text{eval}}(z)}[-\log p_n(z) + \log p_{\text{eval}}(z)] = \text{KL}[p_{\text{eval}}(z) \,\|\, p_n(z)].$$

Now starting from Equation 5, it can also be written as

$$d(p_n, p_{\text{eval}}) = -\mathbb{E}_{p_{\text{eval}}(z)}[\log p_n(z)] - \text{H}[p_{\text{eval}}(z)] = \text{H}[p_{\text{eval}}(z) \,\|\, p_n(z)] - \text{H}[p_{\text{eval}}(z)].$$

Equating these two expressions for the discrepancy leads to a standard information-theoretic decomposition:

$$\underbrace{\text{H}[p_{\text{eval}}(z) \,\|\, p_n(z)]}_{\text{cross entropy}} = \underbrace{\text{KL}[p_{\text{eval}}(z) \,\|\, p_n(z)]}_{\text{KL divergence}} + \underbrace{\text{H}[p_{\text{eval}}(z)]}_{\text{entropy}}$$

where the cross entropy measures predictive performance and the entropy measures data dispersion. $\square$

# B Proofs of Propositions 4 to 6

**Proposition 4** *Let $y_{1:m}^+$ and $p_n(y|\theta)p_n(z|\theta)p_n(\theta)$ be a combination of data sequence and generative model that yield $p_n(\theta|y_{1:m}^+) \to \delta_{\theta_\infty}(\theta)$ as $m \to \infty$. Then the expected conditional predictive entropy, $\mathbb{E}_{p_n(\theta)}[\mathrm{H}[p_n(z|\theta)]]$, is a Bayes estimator of $\mathrm{H}[p_n(z|y_{1:\infty}^+)]$, the marginal predictive entropy after a Bayesian update on infinite new data, $y_{1:\infty}^+$.*

*Proof* Since $y_{1:\infty}^+$ recovers a single setting of $\theta$, reasoning about $\theta$ is equivalent to reasoning about $y_{1:\infty}^+$, following the argument presented in Fong et al (2023). This allows us to apply Proposition 1 with $F = \mathrm{H}[p_n(z|y_{1:\infty}^+)]$ and $f(\theta) = \mathrm{H}[p_n(z|\theta)]$, which gives $\eta^* = \mathbb{E}_{p_n(\theta)}[\mathrm{H}[p_n(z|\theta)]]$. $\qquad\square$

**Proposition 5** *Assume the data and model from Proposition 4. Then the expected information gain in the model parameters, $\mathrm{EIG}_\theta$, from observing $z$ is a Bayes estimator of the infinite-step predictive information gain, $\mathrm{IG}_z(y_{1:\infty}^+)$.*

*Proof* The information gain to be estimated, $\mathrm{IG}_z(y_{1:\infty}^+)$, is defined as the reduction in the model's predictive entropy from a Bayesian update on infinite new data, $y_{1:\infty}^+$:

$$\mathrm{IG}_z(y_{1:\infty}^+) = \mathrm{H}[p_n(z)] - \mathrm{H}[p_n(z|y_{1:\infty}^+)].$$

Combining the known $\mathrm{H}[p_n(z)]$ with the Bayes estimator of $\mathrm{H}[p_n(z|y_{1:\infty}^+)]$ from Proposition 4 gives

$$\mathrm{EIG}_\theta = \mathrm{H}[p_n(z)] - \mathbb{E}_{p_n(\theta)}[\mathrm{H}[p_n(z|\theta)]]$$

as a Bayes estimator of $\mathrm{IG}_z(y_{1:\infty}^+)$. $\qquad\square$

**Proposition 6** *Let $z = y_{n+1}$. Assume $y_{1:n}$ are independent and identically distributed, with $y_i \sim p_{\mathrm{train}}(y)$, and assume $p_n(z) = \mathbb{E}_{p_n(\theta)}[p_n(z|\theta)]$ is a model intended to directly approximate $p_{\mathrm{train}}(z)$. Then the expected information gain in the model parameters, $\mathrm{EIG}_\theta$, from observing $z$ is a Bayes estimator of the true one-step expected information gain, $\mathrm{EIG}_\theta^{\mathrm{true}}$, where the expectation is with respect to $p_{\mathrm{train}}(z)$.*

*Proof* The expected information gain to be estimated, $\mathrm{EIG}_\theta^{\mathrm{true}}$, is defined as the reduction in the model's parameter entropy from a Bayesian update on new data, $z$, where $z$ is drawn from $p_{\mathrm{train}}(z)$:

$$\mathrm{EIG}_\theta^{\mathrm{true}} = \mathrm{H}[p_n(\theta)] - \mathbb{E}_{p_{\mathrm{train}}(z)}[\mathrm{H}[p_n(\theta|z)]].$$

The second term here can be estimated by applying Proposition 1 with $F = \mathbb{E}_{p_{\mathrm{train}}(z)}[\mathrm{H}[p_n(\theta|z_{1:m})]]$ and $f(\theta) = \mathbb{E}_{p_n(z|\theta)}[\mathrm{H}[p_n(\theta|z)]]$. The Bayes estimator that results from this, $\eta^* = \mathbb{E}_{p_n(z)}[\mathrm{H}[p_n(\theta|z)]]$, can be combined with the known current entropy, $\mathrm{H}[p_n(\theta)]$, to produce

$$\mathrm{EIG}_\theta = \mathrm{H}[p_n(\theta)] - \mathbb{E}_{p_n(z)}[\mathrm{H}[p_n(\theta|z)]]$$

as a Bayes estimator of $\mathrm{EIG}_\theta^{\mathrm{true}}$. $\qquad\square$

# C   Implementation details

Code to generate Figures 3 and 4 is available at `github.com/fbickfordsmith/rethinking-aleatoric-epistemic`.

## C.1   Figure 3

We consider predicting an output, $z \in \mathbb{R}$, corresponding to an input, $x \in \mathbb{R}$. The training data, $y_{1:n}$, comprises $n = 2$ input-label pairs: $y_1 = (-2, \tanh(-2))$ and $y_2 = (2, \tanh(2))$. We use this to compute a Gaussian-process predictive posterior, $p_n(z|x) = p(z|x, y_{1:n})$, based on a generative model comprising a Gaussian likelihood function, $p(z|x, \theta) = \text{Normal}(z|\theta(x), \sigma^2)$, where $\sigma = 0.1$, and a Gaussian-process prior, $\theta \sim \text{GP}(0, k)$, where $k(x, x') = \exp(-(x - x')^2/2)$. We compare this with a reference distribution, $p_{\text{eval}}(z|x) = \text{Normal}(y|\tanh(x), \sigma^2)$. Using $p_n(z|x)$ and $p_{\text{eval}}(z|x)$, we compute three quantities for $x \in [-8, 8]$: the predictive uncertainty, $h[p_n(z|x)]$; the data dispersion, $h[p_{\text{eval}}(z|x)]$; and the expected score, $\mathbb{E}_{p_{\text{eval}}(z|x)}[s(p_n, z)]$. We do this for two loss functions: $\ell(a, z) = (a - z)^2$ and $\ell(a, z) = \max(0, z) \cdot (a - z)^2$.

## C.2   Figure 4

We consider two cases of predicting $z = y_{n+1}$: a discrete case and a continuous case. In the discrete case we have $y \in \{0, 1\}$, data generated from $p_{\text{train}}(y) = \text{Bernoulli}(y|\eta = 0.5)$, and the Bayesian generative model is

$$p(y, \eta|\alpha, \beta) = p(y|\eta)p(\eta|\alpha, \beta)$$
$$p(y|\eta) = \text{Bernoulli}(y|\eta)$$
$$p(\eta|\alpha, \beta) = \text{Beta}(\eta|\alpha, \beta).$$

In the continuous case we have $y \in \mathbb{R}$, data generated from $p_{\text{train}}(y) = \text{Normal}(y|\mu = 1, \sigma^2 = 1)$, and the Bayesian generative model is

$$p(y, \mu, \lambda|\alpha, \beta, \kappa, m) = p(y|\mu, \lambda)p(\mu|m, \kappa, \lambda)p(\lambda|\alpha, \beta)$$
$$p(y|\mu, \lambda) = \text{Normal}(y|\mu, \lambda^{-1})$$
$$p(\mu|m, \kappa, \lambda) = \text{Normal}(\mu|m, (\kappa\lambda)^{-1})$$
$$p(\lambda|\alpha, \beta) = \text{Gamma}(\lambda|\alpha, \beta).$$

In both cases we can compute exact Bayesian posteriors (Murphy, 2022), and there is some $n$ for which $p_n(z) = p_{\text{train}}(z)$ (Doob, 1949; Freedman, 1963; 1965). For each case we sample four datasets, $y_{1:n}$, with $y_i \sim p_{\text{train}}(y)$ and $n \in (1, 10, 100, 1000)$. On each dataset we compute the Bayesian parameter posterior, $p_n(\theta) = p(\theta|y_{1:n})$, where $\theta = (\alpha, \beta)$ or $\theta = (\alpha, \beta, \kappa, m)$, and then compute two quadratic estimation errors. The first is the error from approximating the "true" expected information gain in $\theta$ with the standard, model-based expected information gain in $\theta$:

$$\sqrt{\varepsilon_\theta} = \text{EIG}_\theta - \text{EIG}_\theta^{\text{true}}$$
$$= (\text{H}[p_n(\theta)] - \mathbb{E}_{p_n(z)}[\text{H}[p_n(\theta|z)]]) - (\text{H}[p_n(\theta)] - \mathbb{E}_{p_{\text{train}}(z)}[\text{H}[p_n(\theta|z)]])$$
$$= \mathbb{E}_{p_{\text{train}}(z)}[\text{H}[p_n(\theta|z)]] - \mathbb{E}_{p_n(z)}[\text{H}[p_n(\theta|z)]].$$

The second is the error from approximating the infinite-step information gain in $z$ with the expected information gain in $\theta$:

$$\sqrt{\varepsilon_z} = \text{EIG}_\theta - \text{IG}_z(y_{1:\infty}^+)$$
$$= (\text{H}[p_n(z)] - \mathbb{E}_{p_n(\theta)}[\text{H}[p_n(z|\theta)]]) - (\text{H}[p_n(z)] - \text{H}[p_{\text{train}}(z)])$$
$$= \text{H}[p_{\text{train}}(z)] - \mathbb{E}_{p_n(\theta)}[\text{H}[p_n(z|\theta)]].$$

We average over 50 repeats with different random-number seeds.

