# OpenReview forum: "Rethinking Aleatoric and Epistemic Uncertainty"
_ICML.cc/2025/Conference — ICML 2025 poster_

### Official Review · Reviewer_Atnu · 2025-03-10

**Overall Recommendation:** 3

**Summary:**

This paper revisits the concepts of aleatoric and epistemic uncertainty in machine learning. It identifies inconsistencies in how these uncertainties are commonly discussed. The authors argue that traditional definitions are overly simplistic, causing confusion. They propose a decision-theoretic approach to clarify uncertainty, defining it through the expected loss of optimal predictions. This framework allows a clearer distinction between uncertainty that can and cannot be reduced by additional data. Additionally, the authors critique commonly used information-theoretic measures like entropy and BALD scores. They demonstrate these metrics are useful practically, but often inaccurately estimate uncertainty. The paper emphasizes that subjective uncertainty estimates from models should not replace objective external evaluations. Overall, the authors aim to provide a unified, clear foundation for future research on uncertainty estimation.

**Claims And Evidence:**

Examples are supported by simple proofs. Experimental evidence is only anecdotal and limited to BALD.

**Essential References Not Discussed:**

I do not think the following references are strictly essential, but certainly tangential and thus worth a read.

@ "Training data need not be direct examples of the predictive task": This is reminiscent of the notion of "Institutional Separation" introduced in this 2024 ICML paper: https://arxiv.org/abs/2404.04669

@ "Reasoning about new data can (but will not always) yield a unique uncertainty decomposition": The Bayesian selection of new data was discussed here (https://www.jmlr.org/papers/v24/21-1067.html) and here (https://arxiv.org/abs/2406.12560). On a slightly different note, the paper (as far as I could check) focuses on acquiring new data exclusively in the context of active learning. It has recently been noted that the decision-theoretic embedding of data acquisition extends beyond active learning and comprises e.g. self-training in SSL, boosting, and bandits: https://proceedings.neurips.cc/paper_files/paper/2024/hash/0337b41b4e8b2eb5d7ab161ffd42cf3b-Abstract-Conference.html (For an example of Bayes-optimal acquisition in SSL, see https://proceedings.mlr.press/v216/rodemann23a/rodemann23a.pdf)

**Experimental Designs Or Analyses:**

It appears the author did not share code to reproduce experimental evidence, violating open science standards.

**Methods And Evaluation Criteria:**

The paper's conceptual nature implies no comprehensive benchmark study is required. Proofs are clear and easy to follow. Experimental illustration could be enhanced.

**Other Comments Or Suggestions:**

Minor remarks/typos:

- "foundations ideas" →  "foundational ideas"

- "this spurious associations" →  "these spurious associations"

- "implicitly using favouring" → either "implicitly using" or "favouring."

-  "defined to conditional" →  "defined conditional" or "defined to be conditional." ?

- "lead to different reduction" →  "lead to different reductions."

-  "Variously referred in" →  "variously referred to in." ?

- "thus that objective reasoning" → better phrased as "thus objective reasoning."

- "show that recovers" → missing "this," should be "show that this recovers."

-  "down to extremely pure, unambiguous case" → should be plural, "cases."

- "over horizon of m observations" → "over a horizon..."

- "Justifying the need for external grounding." → This sentence fragment should be combined with the preceding sentence.

- "We average over 50 repeats with different random seeds." →  "We average over 50 random seeds."

**Other Strengths And Weaknesses:**

The paper occasionally uses different terms interchangeably without explicitly stating equivalences clearly. For example, "statistical dispersion in data" and "noise" are used interchangeably. It would be clearer to pick one term and consistently use it after an initial definition. I also encourage the authors to clearly note some examples (e.g., variance and entropy derivations) as standard textbook derivations to avoid confusion about novelty.

**Questions For Authors:**

See above.

**Relation To Broader Scientific Literature:**

I welcome a) the decision-theoretic perspective on predictive UQ/model uncertainty and b) the shift away from (mostly purely probabilistic) UQ towards a more comprehensive view on modeling and reasoning in machine learning. Of course, a) allows for b), but I consider both parts of independent interest and valuable contributions on its own. Generally, I like the writing. The presentation is clear and the setup is well-motivated.

Having said this, I identify two main concerns with the current state of the paper. While the first one is a conceptional caveat that might not be resolved so easily, the second refers to the presentation and should be fairly easy to address by minor revisions.

1. As strong as I support the authors in arguing against the simplistic aleatoric-epistemic dichotomy, I am afraid the authors introduce another such (and equally questionable) dichotomy by fundamentally differentiating between objective model evaluation and subjective uncertainty quantification. Let me explain in detail. The authors start by emphasizing the basic rationality axioms by VNM and Ramsey but to my great regret, they fail to expand on them nor explicitly use them. The decision-theoretic embedding would allow for a rigoros study of how the choice of axioms affects the utility (i.e., loss function) in machine learning. Instead, the authors naively equate the predictive task with a loss function. I appreciate how the authors discuss the subjectivity of UQ and expose it to be based on the internal belief state of a model which does not necessarily need to correspond to what the authors call p_{eval}, the law of interest for evaluation. However, they miss the opportunity of doing the same for the choice of the loss function. The authors write, "Bayes optimality simply means we are taking an action that reflects our belief state; it says nothing at all about how well that belief state matches reality." I could not agree more, but the same applies to the loss. For instance, consider the desiderata of a loss function to reflect multiple users' preferences (total orders). It has been long recognised in the social choice literature (e.g. Arrow's famous impossibility result) that no total order exists to aggregate such preferences under reasonable assumptions on the aggregation. See https://apps.dtic.mil/sti/tr/pdf/AD0708563.pdf for more background. In other words, by naively conditioning their decision-theoretic analysis on a real-valued loss (i.e., stipulating a total order), the authors exclude all subjectivity in how such a loss function arises. I consider this to be somewhat inconsistent with the differentiation between p_eval and p_train. Why should a machine learning model be evaluated with respect to different probability distributions but always with respect to the same loss? It was already recognised by Abraham Wald in his seminal 1949 paper "Statistical Decision Functions" (Ann. Math Stat.), that in the case where P_eval is not known and the decision maker has only ordinal preferences, rationality axioms imply a classical maximin instead of the Bayes criterion. Notably, these order theoretic deliberations on the loss aren't purely theoretical. They have been applied in the ML community recently. For instance, https://proceedings.mlr.press/v216/jansen23a/jansen23a.pdf and https://papers.nips.cc/paper_files/paper/2024/hash/b1f140eeee243db24e9e006481b91cf1-Abstract-Conference.html derived optimal procedures for multivariate random variables with locally varying scale of measurement (e.g. cardinal (e.g. autmoatic evaluation of ML model) and ordinal (e.g. human ranking of ML model's output like in RLHF) criteria). In summary, just like "probability does not exist" (de Finetti), i.e., there is no objective probability, there is no objective loss tied to a predictive task. I thus encourage the authors to reconsider the "objective" terminology and reframe their concept as e.g. addressing probabilistic vs. non-probabilistic uncertainty and not subjective vs. objective reasoning. This is further supported by the fact that there workarounds avoiding the sensitivity of Bayes-actions towards the choice of prior. Besides classical objective Bayesian approach (see the cited Berger book), these workarounds comprise "prior near ignorance" credal sets of priors (i.e. convex sets of probability measures that represent partial ignorance), see e.g. https://alessiobenavoli.com/research/prior-near-ignorance/

2. The paper could benefit from some more constructive and practical recommendations for ML practitioners. This is not my area of expertise, but I feel the paper's accessibility to the (applied) ML community could be improved by e.g. a more tangible case study. Why not expand the experiments on BALD to a more comprehensive setup involving both data acquisition and prediction?

**Theoretical Claims:**

Yes, the proofs are correct as far as I could check. This comes at little suprise, as the claims are mostly based on previous work and well-known results.

---

> ### Author Rebuttal · Authors · 2025-04-01
>
> Thank you for your review.
>
> We are pleased to see positive feedback on multiple aspects of the paper:
>
> 1. Strong motivation
> 2. Interesting use of decision theory
> 3. Comprehensive view on reasoning and learning
> 4. Clear writing, proofs and overall presentation
>
> We also appreciate your thoughtful, in-depth conceptual comments along with useful pointers for lower-level improvements.
>
> ### **Loss functions**
>
> > I am afraid the authors introduce another such (and equally questionable) dichotomy by fundamentally differentiating between objective model evaluation and subjective uncertainty quantification.
> >
>
> You raise an astute point that is more aligned with our perspective than you might think.
>
> While we agree that a loss function is typically an imprecise abstraction of our internal desires for how a system should behave (and thus there is a subjective element in its construction), we argue that this falls under our problem definition because it relates to our definition of optimality in an ideal world, and not how that optimality is achieved. Core to our arguments is the idea that all uncertainty characterisation should start from the problem definition; our technical contributions are then based on how this can be done in a rigorous way. This results in a goal-driven notion of uncertainty that is rigorous given a problem definition (note that this setup is very standard across machine learning and not just the uncertainty-quantification literature).
>
> Another key point is that even if we are unsure about how best to set up our loss function, this is not an uncertainty that arises from a *lack of information*. There is therefore no notion of changes in uncertainty from having access to more data or knowing the underlying data-generating process, which is ultimately what we wish to characterise in our decompositions. The choice of loss function is thus objective from a statistical-terminology perspective, even if it is a subjective decision (in the lay sense) in practice. We believe this use of terminology reflects a long history within decision theory, perhaps most prominently in the work of Savage (1971), who linked subjective beliefs to externally grounded evaluations through scoring rules.
>
> Given the above, we do not feel it is problematic to condition our analysis on a real-valued loss, but we do agree that the importance and “subjectivity” of choosing the loss function warrants more discussion in the paper, and we will happily add this in our update.
>
> ### **Practical implications**
>
> > The paper could benefit from some more constructive and practical recommendations for ML practitioners.
> >
>
> Great point. We have laid out some points in “Practical implications” in our response to Reviewer MLbH.
>
> ### **Case study**
>
> > the paper's accessibility to the (applied) ML community could be improved by e.g. a more tangible case study
> >
>
> This is an excellent idea.
>
> In our code repo (link below) we have added a new practical demonstration of some of the key ideas from the paper. We look at Gaussian-process regression and show how the loss function affects three things:
>
> 1. The model’s subjective uncertainty, $h_\ell[p_n(z)]$
> 2. The model’s discrepancy, $d(p_n, p_\mathrm{eval})$, with respect to a reference system
> 3. Which data gets prioritised during data acquisition
>
> We also highlight the crucial distinction between model-based uncertainty quantification and externally grounded evaluation using a scoring rule or discrepancy function.
>
> We do this by comparing a standard quadratic loss against a weighted quadratic loss that encodes a preference for accuracy on larger values of $z$ ($z$ might be a variable we need to be large, such as the solubility of a candidate drug molecule).
>
> We plan to add some of these plots to the paper, along discussion relating it to practical applications.
>
> ### **Code**
>
> > It appears the author did not share code
> >
>
> Thank you for flagging this oversight. We have made our code available at https://anonymous.4open.science/r/rethinking-aleatoric-epistemic-00DF.
>
> ### **Experimental evidence**
>
> > Experimental evidence is only anecdotal and limited to BALD.
> >
>
> We recognise that our experiment is simple, but we contest the characterisation of our evidence as anecdotal. As explained in the paper, our experiment is carefully chosen to support our empirical claim, complementing results from past work.
>
> ### **Existing results**
>
> > I also encourage the authors to clearly note some examples… as standard textbook derivations
> >
>
> Good point. We will flag existing results clearly.
>
> ### **Terminology**
>
> > The paper occasionally uses different terms interchangeably without explicitly stating equivalences clearly.
> >
>
> This is useful feedback. We will standardise the terminology.
>
> ### **Related work**
>
> Thanks for these pointers. We will happily add citations.
>
> ### **Typos**
>
> We appreciate you pointing these out. We will fix them.

---

> > ### Comment · Reviewer_Atnu · 2025-04-01
> >
> > Thanks for your detailed reply. I greatly appreciate it! I feel most of my points are sufficiently addressed - however, my main point is not among them. I am *not* convinced by the way you motivate your fundamental distinction between objective model evaluation and subjective uncertainty quantification. Reviewer MLbH appears to have a hard time accepting this strict dichotomy, too. After considering your reply, I am still under the impression this distinction is artificial and not rigorously grounded in decision theory.
> >
> > > Because it relates to our definition of optimality in an ideal world, and not how that optimality is achieved
> >
> > If that is your main motivation for distinguishing between objective model evaluation and subjective uncertainty quantification, it is a pretty vague one. Can the authors explain why the choice of the loss function is not concerned with "how that optimality is achieved"? A risk functional (mapping from the functional space of loss functions) clearly determines how optimality is achieved, yes, but so does the loss function (and crucially, its underlying domain).
> >
> >
> > > We believe this use of terminology reflects a long history within decision theory
> >
> > This is clearly wrong. It has been remarked by none other than the founding father of statistical decision theory himself (Abraham Wald) that for non-cardinal preferences, basic rationality axioms imply maximin strategies instead of Bayes criteria, on which the authors appear to base their whole argument. The classic Savage paper deals with elicitating personal probability distributions generally. This does not mean elicitation of order preferences on the loss domain is disregarded in decision theory. Quite the contrary, I would argue that a substantial part of decision theorists even work on order theory directly, or at least consider the basis of decision theory.
> >
> > I would like to emphasize that my concerns regarding the order structure implied by the loss function's domain are not purely theoretical. There are strong impossibility results if the loss shall simply represent more than one subject's preferences. Any collective agreement on the loss' underlying order is affected, rendering these limitations relevant to e.g. democratizing AI, see https://arxiv.org/pdf/2206.02786 for instance.
> >
> > EDIT: All in all, I strongly encourage the authors to discuss the loss function's subjective elements (even if tied to an objective problem). After reading the paper again, I do not think my concerns are unresolvable. But the author's simplified abstraction (just like any) has limitations that should be discussed in the revised version of the paper.

---

> > > ### Author Response · Authors · 2025-04-05
> > >
> > > > All in all, I strongly encourage the authors to discuss the loss function's subjective elements (even if tied to an objective problem). After reading the paper again, I do not think my concerns are unresolvable. But the author's simplified abstraction (just like any) has limitations that should be discussed in the revised version of the paper.
> > > >
> > >
> > > Thank you for your continued engagement. Your input is really helping shape the updates we will make to the paper. It is important to us that we get this right.
> > >
> > > We believe we are aligned with you on three central points:
> > >
> > > 1. **There are cases (eg, group decisions) where preferences do not imply a loss function.** Our assumption of a loss function means we cannot cover these cases.
> > > 2. **Even if there is a well-defined notion of loss, it will vary from one decision-maker to another.** Nothing in our analysis is objective in the sense of applying to all decision-makers.
> > > 3. **There are multiple approaches to decision-making.** We use expected-loss minimisation to produce a coherent synthesis of ideas that recovers widely used quantities. This does not mean using expected loss is the only possible approach; alternatives include minimax.
> > >
> > > Point 1 is a limitation we are comfortable with but do not by any means want to ignore. We will clarify the scope of our analysis as it stands and highlight the potential for future work exploring the difficulties you rightly point out.
> > >
> > > Point 2 is something we are keen to communicate more clearly. We understand your issue with “subjective reasoning vs objective reasoning”, and we are planning to change our terminology. One option is “internal reasoning vs external reasoning”; another is “subjective Bayesian reasoning vs frequentist evaluation”. Let us know if you have thoughts on this.
> > >
> > > Point 3 is also something we are happy to provide more context on. This relates to a remaining issue that you raised:
> > >
> > > > Can the authors explain why the choice of the loss function is not concerned with "how that optimality is achieved"?
> > > >
> > >
> > > Our point here was to emphasise that even if we assume we have a loss function (ie, “our definition of optimality”), we still need to choose a higher-level procedure for making decisions (ie, “how that optimality is achieved”), such as expected-loss minimisation or minimax.
> > >
> > > While we will be unable (due to ICML restrictions) to respond to any further comments you have here, we will take seriously any remaining concerns when we update the paper.

---

### Official Review · Reviewer_43ee · 2025-03-11

**Overall Recommendation:** 3

**Summary:**

This paper argues that the current view on the decomposition of uncertainty into (reducible) epistemic and (non-reducible) aleatoric uncertainty is not only insufficient but also inappropriate from the theoretical viewpoint. They argue that the whole notion of predictive uncertainty should be grounded in a loss function over actions, entering into an argument of rational behavior.

**Claims And Evidence:**

I would argue that this is one of the few papers I have seen that does not make a technical but rather a didactical claim: That they provide clarity on three fronts: (i) that a loss function drives the principled treatment of uncertainty, (ii) that decomposition of uncertainty is usually not possible, and (iii) that model-based uncertainty can be interpreted as estimating the model predictive performance on unseen data, which is however not a theoretical substitute for external grounding. Even though I find the whole work compelling and quite well formalized, I am inclined to say that the paper somewhat falls short in its goal to bring clarity. The paper is excellently written, but it is extremely dense and technical with a lot of notation, which is understandable and necessary, but it makes the paper quite difficult to read, and in many parts, it is necessary to read several times and even then it does not become entirely clear (see also my questions below). I think that this paper would need at least one additional page but probably more to give justice to its own ambition of really bringing more clarity. This way, the paper above all raises awareness but, at least in my case, not really much clarity.

**Essential References Not Discussed:**

None I am aware of

**Experimental Designs Or Analyses:**

There is a minimalistic experimental design around BALD for illustrative purposes, which is nice and would also be necessary in other parts of the paper.

**Methods And Evaluation Criteria:**

n/a (interestingly)

**Other Comments Or Suggestions:**

This paper is really different from standard papers one reviews at ICML since there is no technical method, no result tables, bold figures etc. Personally I find this very refreshing, but it is problematic even only fairly review this paper into the new ICML review scheme, which is entirely oriented towards papers with tables with bold numbers.  I appreciate the attempt to try something different. Frankly I also have no suggestion of what to take out from the paper since everything appears relevant; there seems just to be too little space available to properly convey the message.

**Other Strengths And Weaknesses:**

None

**Questions For Authors:**

- line 149 left: Why would the Shannon entropy of the training data be n log 2?
- I am not sure what to think of this type of issue that "... is only an estimator of [the true uncertainty quantity]". I mean the authors surely will agree that we will always only be able to estimate the true uncertainty, whichever it is, at least in what *you* here define as aleatoric and epistemic uncertainty. So in the light of an impossibly perfect estimate, what would be even the desireable objective in your viewpoint?
- I am also not sure that I follow the argument of unique decomposability of the expected uncertainty reduction only if the additional training samples are determined without any noise (bottom right of page 5). I don't see what is wrong about a decomposition that still involves the expected value of those additional data points. At least this part doesn't become clear to me.

**Relation To Broader Scientific Literature:**

Excellent

**Theoretical Claims:**

There are some few propositions, most of which I checked the proofs and found them reasonable. These propositions are cool and should be there. They are not a game changer for the paper in either direction though, and I wouldn't even call them "claims".

---

> ### Author Rebuttal · Authors · 2025-04-01
>
> Thank you for your review.
>
> We are happy to see you highlight a number of positive aspects of the paper:
>
> 1. Clear writing
> 2. Reasonable theory
> 3. Useful experimental results
> 4. Clear contextualisation within the literature
> 5. Refreshing style of contribution
>
> We also recognise there are some things you think should be improved.
>
> ### **Clarity and space**
>
> > Even though I find the whole work compelling and quite well formalized, I am inclined to say that the paper somewhat falls short in its goal to bring clarity. The paper is excellently written, but it is extremely dense and technical with a lot of notation, which is understandable and necessary, but it makes the paper quite difficult to read
> >
>
> We agree that the paper is necessarily idea-dense in order to communicate the full picture with appropriate nuance. Happily we think the paper’s clarity could quite easily be improved using the extra page allowed in the camera-ready paper, some careful rearrangement of the content, and extra efforts to highlight the key takeaways.
>
> A key improvement we can make is to provide visual examples of the points we are making (see “Case study” in our response to Reviewer Atnu). Another thing we will try is presenting Examples A1-A3 and B1-B3 together: then the logic of the main technical content will be uninterrupted and the progression across the examples will be clearer. Finally we will make more use of the appendix to provide more verbose explanations of the trickier concepts covered in the paper.
>
> ### **Paper style**
>
> > This paper is really different from standard papers… Personally I find this very refreshing, but it is problematic even only fairly review this paper into the new ICML review scheme…
> >
>
> It is great to hear that you find our contribution refreshing. We believe it could provide real value to the community, especially after we update for improved clarity based on your feedback. As noted above, we do feel there should be sufficient space to convey our message more clearly.
>
> ### **Imperfect estimators**
>
> > I am not sure what to think of this type of issue that "... is only an estimator of [the true uncertainty quantity]".
> >
>
> You are right: estimation is what we have to do in practice. Two points are worth highlighting.
>
> First, past work has often discussed common estimators as if they are the quantities they are in fact only approximating. We think emphasising the potential inaccuracy of the estimators has value in resolving misconceptions and supporting more clear-eyed use of common quantities.
>
> Second, by establishing that these are only estimators, our work reveals the scope for alternative estimators that might be superior. For example, it might be clear that the infinite-step irreducible predictive uncertainty should be effectively zero, which a practitioner can use directly; Figure 2 in https://arxiv.org/abs/2404.17249 shows this can be better than using standard estimators. From a more theoretical perspective, an important implication of Propositions 1-4 is that the standard uncertainty estimators we consider are only optimal if we use a quadratic estimation loss, with other losses yielding different optimal estimators. Our exposition provides a template for deriving better estimators.
>
> ### **Unique uncertainty decomposition**
>
> > I don't see what is wrong about a decomposition that still involves the expected value of those additional data points.
> >
>
> The key point here is that stochasticity in the data makes the decomposition depend on the data-generating process: any notion of irreducible and reducible uncertainty becomes conditional on it. This means we cannot talk about *the* decomposition into irreducible and reducible uncertainty: there are effectively endless possible decompositions because the data-generating process itself depends on design decisions we make (eg, how inputs are sampled or selected). Even if we fix a design policy (https://arxiv.org/abs/1604.08320), the corresponding true data-generating process is still unknown and so we are approximating the true expected decomposition with our model of the data-generating process. We will make updates to clarify this in the paper.
>
> Another thing we would be happy to expand on in the paper (if it is of interest) is the existence of families of distinct data-generating processes that produce the same uncertainty decomposition given a long enough rollout of data generation. For example, it is possible for different experimental-design policies to be equivalent in the extent to which they will reduce uncertainty over a given rollout length.
>
> ### **Training-data entropy**
>
> > line 149 left: Why would the Shannon entropy of the training data be n log 2?
> >
>
> We have $2^n$ possible outcome sequences, $y_{1:n}$, each with probability $1/2^n$. The entropy is
>
> $$
> \mathrm{H}[p_\mathrm{train}(y_{1:n})] = -\sum_{i=1}^{2^n} p_\mathrm{train}(y_{1:n}^{(i)}) \log p_\mathrm{train}(y_{1:n}^{(i)}) = -\sum_{i=1}^{2^n} 2^{-n} \log 2^{-n} = n \log 2.
> $$

---

### Official Review · Reviewer_MLbH · 2025-03-13

**Overall Recommendation:** 2

**Summary:**

The paper examines the concepts of aleatoric and epistemic uncertainty. It highlights inconsistencies in existing discussions of these concepts, attributing them to the limited expressiveness of the aleatoric-epistemic framework in capturing the diverse uncertainty quantities. To address this, the authors propose a decision-theoretic perspective on prediction, deriving formal definitions of model-based uncertainty and statistical dispersion in data. This new framework aims to provide a clearer foundation for future discourse in the field. Additionally, the paper investigates popular information-theoretic quantities, revealing their limitations as estimators of intended metrics while demonstrating their potential utility in guiding data acquisition processes.

**Claims And Evidence:**

Yes. The authors provide proofs for the examples and propositions. Nevertheless, as I noted in the Weaknesses, I find certain arguments and motivations to be insufficiently persuasive.

**Essential References Not Discussed:**

N/A

**Experimental Designs Or Analyses:**

Yes. I think the experiments are not sufficient. See Weaknesses for details.

**Methods And Evaluation Criteria:**

No.

**Other Comments Or Suggestions:**

N/A

**Other Strengths And Weaknesses:**

Strengths:
1. The existing aleatoric-epistemic framework indeed suffers from conceptual ambiguity, coupled with evident terminological misuse in the design of current methodologies. This constitutes a compelling and worthwhile topic for discussion within the field of uncertainty estimation.

2. I agree with the opinions in Section 5.4. I think the analysis really provides some interesting insights.

Weakness:
1. Some of the article’s arguments are somewhat vague and difficult to comprehend, or not convincing enough. For instance:

Line 126: "This assumption is equivalent to stipulating basic axioms of rationality (Ramsey, 1926; von Neumann & Morgenstern, 1947)."
The connection between the cited references and the preceding context remains unclear. Furthermore, the specific mechanism by which this equivalence is ensured is not elucidated.

Line 212: "Given this, we argue that a principled notion of predictive uncertainty cannot be detached from this loss".
I find this perspective both difficult to comprehend and lacking in credibility. Even after thoroughly reviewing the entire article, I remain unable to fully grasp the rationale and necessity for adopting this viewpoint as the foundation. The authors may need to provide a more comprehensive explanation to justify this position and clarify its significance in the context of the study.

2. I find the presentation in Section 3: Key Concepts confusing. Is the author intending for the concepts in this section to be treated as assumptions that must be satisfied, or as prerequisite knowledge that readers are expected to possess?

3. The practical applicability of this work remains ambiguous. What tangible benefits might arise from replacing the traditional aleatoric-epistemic view with the proposed decision-theoretic perspective at the application level? Could it facilitate more accurate analysis of uncertainty sources? Analyzing the sources of uncertainty is a primary developmental goal of the aleatoric-epistemic framework, and the application value of this direction should be explored in that context. However, the experimental section of the paper is severely lacking, with insufficient empirical validation to support the claims.

4. As I read through Section 5.3 and realized that the authors’ primary contributions concluded at that point, I experienced a slight sense of surprise and disappointment. I had anticipated the emergence of a concrete uncertainty decoupling algorithm guided by the authors’ novel perspective. It is possible that this reflects a limitation in my own understanding, but I currently hold the view that, while this work offers ample and innovative theoretical analysis, it remains incomplete in its present form. I recognize one of the authors’ assertions: that a unique decomposition of uncertainty into reducible and irreducible components is often unattainable. However, I contend that this should not serve as a justification for forgoing the exploration of specific algorithms. Under the traditional aleatoric-epistemic framework, researchers have pursued a variety of uncertainty decoupling algorithms, even if these are not always precise. If the authors aim to substantiate the superiority of their decision-theoretic perspective over conventional approaches, the development of concrete algorithms accompanied by comprehensive experimentation is indispensable.

5. Inappropriate or Inaccurate Literature Citations: (1) Page 1, Line 14: The citation (Kendall & Gal, 2017) merely introduces a variance-based measure for epistemic uncertainty within the related work section. Utilizing it as an example in this context appears somewhat inappropriate. (2) Page 1, Line 18: The reference (van Amersfoort et al., 2020) does not employ distance-based measures to quantify epistemic uncertainty. The study explicitly states that it does not engage in the decoupled analysis of aleatoric and epistemic uncertainty, rendering this citation potentially misleading.

6. Inappropriate Submission Title: The author should focus more precisely on the unique contributions of the paper when selecting the title, rather than adopting a generic phrase such as "Rethinking Aleatoric and Epistemic Uncertainty." This title lacks distinctiveness, as the reconsideration of aleatoric and epistemic uncertainty has been an ongoing topic of discussion for an extended period.

**Questions For Authors:**

See Weaknesses

**Relation To Broader Scientific Literature:**

This study examines the limitations of the traditional aleatoric-epistemic framework and introduces a distinctive decision-theoretic perspective. It further highlights that widely used information-theoretic quantities may serve as inadequate estimators of the constructs they are commonly assumed to measure.

**Theoretical Claims:**

Yes. The article incorporates a substantial number of mathematical expressions; however, it does not involve highly complex derivation or proof processes.

---

> ### Author Rebuttal · Authors · 2025-04-01
>
> Thank you for your review.
>
> Your critical feedback is much appreciated. We hope our responses and new demonstrative plots help alleviate your concerns.
>
> ### **Loss functions and uncertainty**
>
> > "Given this, we argue that a principled notion of predictive uncertainty cannot be detached from this loss"… I remain unable to fully grasp the rationale and necessity for adopting this viewpoint as the foundation.
> >
>
> This is useful feedback—thanks. We will clarify our case.
>
> Our core argument is that notions of uncertainty should be derived from the task we ultimately wish to perform, rather than being chosen in abstract.  The loss is the measure of success on our final task, and our analysis shows that we can then directly derive rigorous notions of uncertainty from this loss using a framework of rational actions, namely minimising the expected loss.
>
> The necessity of adopting this viewpoint as a foundation then manifests in a variety of ways, such as ensuring uncertainties are consistent with rational actions, having an end-goal-driven approach so that uncertainties reflect what we actually care about, and allowing data acquisition to be performed in a meaningful way that targets the problem of interest.
>
> ### **Practical implications**
>
> > The practical applicability of this work remains ambiguous. What tangible benefits might arise…?
> >
>
> Thanks for highlighting this. We will happily add more discussion on the practical benefits. We also plan to add a case study that helps show the implications of our work in a practical setting (see "Case study" in our response to Reviewer Atnu).
>
> One takeaway is to stop using off-the-shelf uncertainty measures and instead derive the one that will be most useful in a given decision problem (eg, data acquisition). We show how to do this.
>
> Another is that we cannot think simply in terms of "Analyzing the sources of uncertainty" in the way the current literature does, with such decompositions themselves being highly subjective in practice. It is hard to characterise the application value of the work in the context of the aleatoric-epistemic viewpoint, as we are ultimately arguing that this is fundamentally flawed.
>
> Other takeaways relate to estimation of uncertainty in practice. Part of this is promoting caution in using common estimators; part is encouraging alternatives. For example, standard estimators can be so inaccurate that we are better off directly using a numerical estimate based on our prior knowledge (see "Imperfect estimators" in our response to Reviewer 43ee).
>
> ### **Concrete algorithms and empirical evidence**
>
> > If the authors aim to substantiate the superiority of their decision-theoretic perspective over conventional approaches, the development of concrete algorithms accompanied by comprehensive experimentation is indispensable
>
> > the experimental section of the paper is severely lacking, with insufficient empirical validation to support the claims
>
> We feel that this misses the primary contribution of our work, which is not algorithmic but instead in showing the inconsistencies and flaws in the current foundations on which uncertainty-quantification algorithms are usually based, as well as providing a more rigorous and principled foundation.  As noted by other reviewers, this makes it an unusual paper, but we do not believe every paper should be about proposing a new algorithm.
>
> By extension, it is not clear what empirical experimentation it would actually make sense to add to the paper, beyond confirming the one empirical claim we make about how the BALD estimator should be understood. We do not agree that there claims being made that are lacking empirical validation, but if there are any experiments you think are missing we will do our best to add them.
>
> To try and provide more clarity on how the work can be used for concrete algorithms, we have also added a new case study in our code base (see "Case study" in our response to Reviewer Atnu). This shows how using a non-standard loss can significantly impact how the uncertainty should be measured.
>
> ### **Title**
>
> > This title lacks distinctiveness
> >
>
> We believe the broad span and foundational nature of our work actually requires a title like the one we use. Our paper brings together ideas from much of the ongoing discussions of aleatoric and epistemic uncertainty in the literature and aims to change the way people think about these concepts themselves.
>
> ### **Key concepts**
>
> > Is the author intending for the concepts in this section to be treated as assumptions that must be satisfied, or as prerequisite knowledge that readers are expected to possess?
> >
>
> The concepts in the section are intended to be didactic rather than assumptions we are making.  The section is also covering some of our problem formulation and synthesising relevant facts from past work, but its core is to introduce the key ideas that underpin our formulations.
>
> ### **Citations**
>
> We will revisit all of the citations you raise.

---

### Official Review · Reviewer_GCxo · 2025-03-14

**Overall Recommendation:** 3

**Summary:**

This paper critiques the concepts of aleatoric and epistemic uncertainty in machine learning predictions, identifying inconsistencies and limitations in existing discussions. The authors argue that the traditional aleatoric-epistemic framework is insufficient to capture all relevant aspects of uncertainty in predictive modeling.

To address these shortcomings, the authors propose a more rigorous framework for model uncertainty based on expected loss values. This approach aims to clarify the concerns discussed and address various aspects of uncertainty more comprehensively, while also capturing existing notions of uncertainty.

The authors provide experimental insights into the BALD score, a popular information-theoretic quantity. Their findings demonstrate that this metric can sometimes measure something substantially different from what it is commonly perceived to quantify (i.e. the infinite-step predictive information gain).

Overall, the paper challenges existing paradigms in uncertainty quantification for machine learning models and proposes a new perspective based on decision theory to address the identified shortcomings.

**Claims And Evidence:**

The paper is carefully written, and the theoretical justifications for the paper's argument are rigorous and make sense.
The experimental results on the performance of the BALD score offer convincing evidence for the authors' claims about the potential misinterpretation of common information-theoretic quantities. The connection between these experiments and Proposition 5 strengthens the paper's argument.

**Essential References Not Discussed:**

The paper's discussion of aleatoric and epistemic uncertainty perspectives shares some themes with this ICLR 2025 paper: https://arxiv.org/abs/2412.18808. In particular, Section C.2 of that paper appears relevant to the decision-theoretic perspective in Section 5.1 here.

**Experimental Designs Or Analyses:**

I liked the experiments, especially in the context of the result of Proposition 5. However, I would have preferred if more explanation of the setup was given in the main body, as I had to consult the appendix to be able to understand Figure 3.

**Methods And Evaluation Criteria:**

The paper's experiments use simple toy models that clearly distill the paper's points and are easy to understand. The BALD score is a popular measure of predictive uncertainty, making it a relevant focus for the experiments.

**Other Comments Or Suggestions:**

- I felt the proofs and statements of propositions 2-5 were a bit too brief, sometimes making the discussion hard to follow. It would be useful to concretely restate the quantities referenced in the proposition statements in mathematical terms.
- Perhaps I missed it, is $EIG_{\theta}$ ever defined in the main body?
- It might be valuable to consider including in the conclusion a brief exploration of any key open questions or potential research directions that this discussion brings to light.
- A number of the proofs (Propositions 2, 3, 5) use the phrase "...follows from the same working as in..." perhaps a more standard word choice would be "reasoning" or "argument" rather than "working"?

**Other Strengths And Weaknesses:**

__Strengths:__
The paper is well-written and provides a valuable clarifying perspective on the commonly held views of epistemic and aleatoric uncertainty in machine learning. It also offers a more grounded alternative approach to understanding and quantifying predictive uncertainty.

__Weaknesses:__
The paper's focus on conceptual discussion, coupled with limited technical results and experiments confined to synthetic data, could potentially position it as more of a position paper, making it better suited to a different venue. However, the message it conveys is likely to be beneficial for the broader machine learning community.

**Questions For Authors:**

I have no specific questions, but please address any comments in my review that may indicate a misunderstanding of the paper's key points.

**Relation To Broader Scientific Literature:**

A large number of works have studied aleatoric/epistemic uncertainty and related notions of uncertainty quantification. This paper both adds to the conversation and connects and clarifies the different perspectives prior works take on this topic, making it a valuable addition to the body of research.

**Theoretical Claims:**

While there are no substantial proofs to verify, there are a few small propositions whose proofs seem sound, and overall the analysis of different perspectives on uncertainty in the literature and their connections to each other is clearly stated and theoretically grounded. Please see my comments below about the clarity of the proofs and their associated proposition statements.

---

> ### Author Rebuttal · Authors · 2025-04-01
>
> Thank you for your review.
>
> We appreciate your positive feedback on a number of points:
>
> 1. Careful writing
> 2. Theoretical rigour
> 3. Convincing empirical evidence
> 4. Clear coverage of prior work
> 5. Potential benefit to the community
>
> We are also grateful that you highlighted some ways to improve the paper.
>
> ### **Paper style**
>
> > The paper's focus on conceptual discussion… could potentially position it as more of a position paper… However, the message it conveys is likely to be beneficial for the broader machine learning community.
> >
>
> We strongly support your emphasis on the benefit the paper could bring to the community, which we believe is ultimately what matters. While we agree that the paper would not have been totally out of place in ICML’s position-paper track, we ultimately felt that the objective and technically precise nature of its core contributions made it a better fit for the standard track, even if it does not fit the common mould within the field, as you note.
>
> ### **Future work**
>
> > It might be valuable to consider including in the conclusion a brief exploration of any key open questions or potential research directions that this discussion brings to light.
> >
>
> Great idea. We will happily add this.
>
> One exciting direction is deriving new problem-driven data-acquisition objectives using the decision-theoretic approach we demonstrate. In particular, we think our work lays foundations for developing loss-calibrated active-learning methods, which are underappreciated in the literature.
>
> A key open question is how reducible uncertainties should best be estimated. Propositions 1-4 reveal existing estimators to be optimal only if we use a quadratic estimation loss, with further work required to establish optimal estimation strategies in other contexts.
>
> ### **Experimental setup**
>
> > I would have preferred if more explanation of the setup was given in the main body, as I had to consult the appendix to be able to understand Figure 3.
> >
>
> This is useful feedback—thanks. We will happily revisit it.
>
> ### **Related paper**
>
> > The paper's discussion of aleatoric and epistemic uncertainty perspectives shares some themes with this ICLR 2025 paper: https://arxiv.org/abs/2412.18808. In particular, Section C.2 of that paper appears relevant to the decision-theoretic perspective in Section 5.1 here.
> >
>
> Thanks for drawing our attention to this work. We agree that it is relevant and will cite it.
>
> ### **Wording of propositions and proofs**
>
> > I felt the proofs and statements of propositions 2-5 were a bit too brief, sometimes making the discussion hard to follow. It would be useful to concretely restate the quantities referenced in the proposition statements in mathematical terms.
> >
>
> > A number of the proofs (Propositions 2, 3, 5) use the phrase "...follows from the same working as in..." perhaps a more standard word choice would be "reasoning" or "argument" rather than "working"?
> >
>
> We appreciate the feedback. We will update the wording and mathematical statement of all propositions and proofs with a view to making things easier to follow and using standard language.
>
> ### **Expected information gain**
>
> > Perhaps I missed it, is $EIG_{\theta}$ ever defined in the main body?
> >
>
> Thanks for flagging this. It is the expected information gain in $\theta$, where $\theta$ represents stochastic model parameters, which is the same thing as the BALD score we refer to throughout the paper. We will take care to clarify this.

---

### Decision · Program_Chairs · 2025-05-01

**Decision:**

Accept (poster)

**Comment:**

This paper argues that the epistemic and aleatoric uncertainty dichotomy, as codified in the existing uncertainty quantification literature, is "insufficiently expressive" and incoherent. The authors instead argue for an alternative task-oriented perspective rooted in Bayesian decision theory. They discuss what tasks yield uncertainty decompositions and the effects of finite data on such uncertainty estimates. The authors finally analyze the popular BALD uncertainty decomposition as a motivating example.

This paper is an unconventional ICML submission. Though the argument is rigorous and well rooted in decision-theoretic principles, it is mainly philosophical, as the theory is limited to propositions, and there is little experimental evidence. Moreover, there are no "practical" takeaways for researchers or practitioners, as the paper is essentially a negative result. Nevertheless, the argument presented in this paper is important and will be of high interest to the uncertainty quantification community. Because the paper's argument is well-grounded in statistical principles (and not merely a subjective argument like a position paper) I recommend acceptance into the main ICML track.

For the camera ready, I suggest that the authors address the minor clarity issues brought up during the review process.